# Characterization of international partnerships in global retinoblastoma care and research: A network analysis

Hannah Girdler[1], Kaitlyn Flegg[2], John Prochaska[3], Helen Dimaras[2,4,5,6,7]*

1 Human Biology Program, Faculty of Arts & Science, University of Toronto, Toronto, Canada, 2 Department of Ophthalmology and Vision Sciences, The Hospital for Sick Children, Toronto Canada, 3 Department of Preventive Medicine & Community Health, University of Texas Medical Branch at Galveston, Galveston, Texas, United States of America, 4 Department of Ophthalmology and Vision Sciences, Faculty of Medicine, University of Toronto, Toronto, Canada, 5 Division of Clinical Public Health, Dalla Lana School of Public Health, University of Toronto, Toronto, Canada, 6 Child Health Evaluative Sciences Program, SickKids Research Institute, Toronto, Canada, 7 Center for Global Child Health, SickKids Research Institute, Toronto, Canada

* helen.dimaras@sickkids.ca, helen.dimaras@utoronto.ca

**Data Availability Statement:** The dataset used in this analysis is included as a Supporting Information file.

## Abstract

Global cooperation is an integral component of global health research and practice. One Retinoblastoma World (1RBW) is a cooperative network of global treatment centers that care for children affected by retinoblastoma. The study aimed to determine the number, scope and nature of collaborations within 1RBW, and uncover how they are perceived to contribute towards improving retinoblastoma outcomes. A cross-sectional, mixed-methods egocentric network analysis was conducted. Treatment centers (n = 170) were invited to complete an electronic survey to identify collaborative activities between their institution (ego), and respective partners (alters). Network maps were generated to visualize connectivity. Key informants (n = 18) participated in semi-structured interviews to add details about the reported collaborations. Interviews were analysed through inductive thematic analysis. Surveys were completed by 56/170 (33%) of 1RBW treatment centers. Collectively, they identified 112 unique alters (80 treatment centers; 32 other organizations) for a total network size of 168 nodes. Most collaborations involved patient referrals, consultations and twinning/capacity building. Interviews identified four main themes: conceptualization of partnership; primary motivation for collaborations; common challenges to collaboration; and benefits to partnership. There is extensive global collaboration to reduce global retinoblastoma mortality, but there is room to expand connectivity through active efforts to include actors located at network peripheries.

## Introduction

By definition, global health practice necessitates global cooperation [1], yet the mechanisms by which global partnerships are structured and operate to achieve their outcomes remain elusive.

**Funding:** We acknowledge the SickKids Ophthalmology Research Fund for funding the study. Author HG was supported by the Queen Elizabeth II Diamond Jubilee Foundation Scholars Program. The funders had no role in study design, data collection and analysis, decision to publish, or preparation of the manuscript.

**Competing interests:** The authors have no competing interests to disclose.

One Retinoblastoma World (1RBW) is a global network aiming to promote optimal care for children with the rare childhood eye cancer retinoblastoma [2, 3]. The highest burden of retinoblastoma is observed in low-and-middle income countries (LMICs) (89% of worldwide cases) where mortality can reach 70% [3]. Delayed diagnosis, limited availability of and access to quality care, and poor compliance with treatment, combined with social, cultural and economic barriers faced by vulnerable communities, are some of the multitude of factors that intersect to result in poor survival [3]. Yet retinoblastoma is curable, especially when detected early and evidence-based care is available and accessible [3]. For this reason, the World Health Organization's Global Initiative for Childhood Cancer emphasizes retinoblastoma as one of the most urgent cancers to address [4].

Treatment centers that make up 1RBW are listed on a publically available map (www.1rbw.org) that features resources and expertise available for care worldwide, as well as country-level incidence estimates. Several international collaborations have shown progress towards improving retinoblastoma outcomes; activities range from implementing an awareness campaign to achieve early diagnosis [5], establishing shared care models [6], standard clinical protocols [7–9] or twinning programs [10–12] to improve health service delivery capacity, using telemedicine to access centralized expertise [13], and development of a multidisciplinary national retinoblastoma strategy which includes a combination of approaches [14]. Studying the dynamics of collaborations can reveal how they work to achieve their intended outcomes. A comprehensive and complete understanding of key global actors, activities and flow of resources, knowledge or other benefits, can help identify ways to strengthen collaborative activities, and promote inclusion of marginalized groups. Social network analyses in health services research can uncover this type of information [15], and have proven insightful in the study of activities related to ophthalmology [16, 17] and cancer [18, 19] care. Therefore, we sought to determine the number, scope and nature of collaborations within 1RBW, and uncover how they are perceived to contribute towards improving global retinoblastoma survival.

## Methods

### Study design

This was a cross-sectional, mixed methods egocentric social network analysis [20], consisting of a survey (quantitative) and semi-structured interview (qualitative) aiming to identify the number and nature of partnerships in 1RBW. The study methodology was aligned with the 3 main stages of social network analysis for health services research: (i) define actors and members in the network; (ii) define relationships between actors, and (iii) analyse the structure of the system [15]. This study was reported as per the Strengthening the Reporting of Observational Studies in Epidemiology (STROBE) guideline for cross-sectional studies (S1 File). Research Ethics Board approval was obtained from The Hospital for Sick Children (Toronto, Canada). Participants completed an electronic informed consent form. Research was conducted in accordance with the Declaration of Helsinki.

### Social network analysis

**(i) Definition of actors & members in the network.** *Description of 1RBW.* The analysis was focused on the 1RBW network (www.1rbw.org), which describes the global retinoblastoma burden and resources available at treatment centers around the world. The centers listed on www.1rbw.org were initially identified by searching relevant literature (e.g. publications, grey literature) and membership lists of professional organizations related to retinoblastoma, ocular oncology, or pediatric oncology. Physician representatives of each identified center

were invited to self-report their data; participation was voluntary. Additionally, centers not identified through this approach were able to make a request to be included in the database. Though 1RBW does not represent an exhaustive list of all centers that manage retinoblastoma patients, it is estimated to represent the vast majority. At the time of the study, there were 170 centers listed on www.1rbw.org.

*Recruitment of stakeholders within 1RBW*. Participants were eligible for the study if they were a healthcare professional (e.g. oncologist, ophthalmologist) leading a retinoblastoma team at a treatment center listed on www.1rbw.org. An 'ego' was defined as any single retinoblastoma treatment center depicted on the www.1rbw.org map which participated in the study. An 'alter' was defined as a collaborator reported by any ego. Collectively, egos and alters were referred to as 'nodes'.

Personalized emails were sent to invite participants to log into their treatment center's 1RBW account, where they would be prompted to provide informed electronic consent and access the survey. Recruitment occurred between May 2016 and August 2017. Only one survey could be submitted per treatment center, and participants were encouraged to consult with their team before finalizing submission. E-mail and telephone reminders were employed to increase response rate. Participants who faced difficulty using the online platform were assisted by phone or Skype to answer questions verbally, and a study team member input their data online.

*Stakeholders outside of 1RBW*. Egos were asked to identify alters, representing any treatment center (within or outside 1RBW) or 'other' organization (i.e. not-for-profit, patient group, government agency, research institute) with which they had collaborated on a retinoblastoma-specific project.

*Author positionality with respect to 1RBW*. All authors work in leading academic centers located in large cities of high-income, English-speaking countries. One author (JP) is a public health scientist who had no connection to 1RBW or the field of retinoblastoma prior to joining the study team. Three authors (HG, KF, HD) are affiliated with a large tertiary retinoblastoma center located in North America; the center is listed on 1RBW and engages in international partnerships. At the time of the study, HG and KF were new to retinoblastoma research. However, HD is a scientist who leads a retinoblastoma research program with a strong focus on global health. In particular, she established www.1rbw.org, and her research team maintains the website, with the express purpose of promoting connection and collaboration within the field. As such, HD acknowledges that these views may have shaped the interpretation of the results.

**(ii) Definition of relationships among actors in the network.** *Network survey*. To define the nature of the relationships among actors, egos completed a survey (S2 File); the questions were modeled after the network analysis survey used for the Brazos Valley Health Partnership [21]. Egos specified the type of activity/interaction (i.e. patient referrals, research projects, patient consultations, twinning/capacity building projects, joint planning) with each alter, along with directionality of each connection (i.e. outdegree = initiated by the ego, indegree = initiated by an alter). Additional questions addressed the nature of the partnership(s) (e.g. formal or informal), and the duration and frequency of activities.

*Semi-structured interviews*. Key informants were selected from survey respondents and invited via email to participate in a semi-structured interview. Key informants were chosen using the maximum variation method; to do this we reviewed responses to the network questionnaire and invited respondents who represented different geographical regions and had diverse experiences with number and types of activities (as described above) or partners (i.e. hospitals, other organizations, local or international, etc.). An interview guide was developed with open-ended questions asking about the history of individual partnerships with alters,

level of engagement, perceived power relationships, financial resources, motivations, and perceived impact of the partnerships (S3 File). Key informants were asked to comment on the duration, intensity, intimacy and reciprocity of their connection(s). Owing to the semi-structured nature of the interview, questions were modified to match the qualities of the ego and partnership history they described. For example, a key informant with only one connection would not be asked to describe their strongest and weakest partnerships (Q2a and Q3a in S3 File), but instead asked to describe that connection as either strong or weak; subsequent questions would be modified accordingly.

The interviews were conducted in English and occurred via telephone, WhatsApp or Skype and were audio recorded. The interviewer (HG) acted according to guidelines for network evaluators [22]. Briefly, they listened actively; recognized patterns and synthesized information; commented on and observed power dynamics and their impacts; identified different forms of leadership; and encouraged participants to tell their stories.

**(iii) Analysis of the structure of the network.** Survey data were exported into Microsoft Excel, stripped of all identifying information and labeled with the unique study ID. The size (i.e. number of alters) and range of connections (i.e. types of alters) present in each ego network were identified. Reciprocity of relationships between egos and alters was assumed. Processed data was imported into Ucinet 6.1358, a tool for network data analysis. Network maps were created using network mapping software (NetDraw 2.41, software for data visualization). Nodes were categorized by World Bank geographical region and country income status [23]. Measures of degree centrality (the immediate number of connections a given organization has to others) and betweenness centrality (a measure of positional importance of an organization within the network as being along pathway among other, otherwise potentially unconnected, organizations) were calculated to quantify organizational connectedness and positions within the various categories of collaboration [24].

Network density, defined as the interconnection between nodes (using a scale of 0% to 100%, where 0% illustrates that there no connections between nodes, and 100% means every node is connected), was calculated for all collaborations and separately for each category of collaboration. Total number of possible collaborations was calculated by the formula $(n)*(n-1)/2$, where n = (egos + unique alters).

Interview recordings were transcribed verbatim. Thematic analysis was conducted following an inductive, explicit, critical realist approach. Data was managed and coded using QSR NVivo 11 software. As analysis progressed, codes emerged and the coding frame was expanded or adjusted through discussion among study team members (HG, HD, KH). Using constant comparative methods, data was compared and contrasted, and categories were grouped and regrouped to represent higher levels of abstraction until major themes emerged.

## Results

### Participants

Fifty-six egos completed the survey (56/170, 33% response rate). Most were located in East Asia and the Pacific (21/56, 38%) followed by Europe and Central Asia, North America and Sub-Saharan Africa, (8/56, 14% each) (Table 1). Egos represented treatment centers in 27 countries (S4 File). All egos were secondary or tertiary facilities and only 2 were located in rural areas; this is consistent with the overall representation in 1RBW, as the majority of facilities able to treat retinoblastoma are located in urban areas, and primary facilities are able to do little more than offer a retinoblastoma diagnosis.

Twenty-three egos were invited to participate in a follow-up semi-structured interview. All accepted to be interviewed, but only 18 were able to schedule interview dates.

**Table 1. Nodes, alters and ties by world bank region.**

| World Bank Region | Nodes | | Alters | | | | | | All Network Members | | Ties (as identified by Nodes) | | | | | | | |
|---|---|---|---|---|---|---|---|---|---|---|---|---|---|---|---|---|---|---|
| | | | Unique Alters | | Unique 'Treatment Center' Alters | | Unique 'Other' Alters | | | | Ties with 'Treatment Center' Alters | | Ties with "Other" Alters | | Total Ties | | Mean # of ties per node |
| | n | % | n | % | n | % | n | % | n | % | n | % | n | % | n | % | |
| East Asia and Pacific | 21 | 13% | 12 | 7% | 11 | 7% | 1 | 1% | 33 | 20% | 51 | 28% | 1 | 1% | 52 | 28% | 2.5 |
| Europe and Central Asia | 8 | 5% | 19 | 11% | 14 | 8% | 5 | 3% | 27 | 16% | 31 | 17% | 5 | 3% | 36 | 20% | 4.5 |
| Latin America and Caribbean | 3 | 2% | 11 | 7% | 9 | 5% | 2 | 1% | 14 | 8% | 5 | 3% | 2 | 1% | 7 | 4% | 2.3 |
| Middle East and North Africa | 3 | 2% | 7 | 4% | 5 | 3% | 2 | 1% | 10 | 6% | 1 | 1% | 2 | 1% | 3 | 2% | 1.0 |
| North America | 8 | 5% | 27 | 16% | 15 | 9% | 12 | 7% | 35 | 21% | 40 | 22% | 13 | 7% | 53 | 29% | 6.6 |
| South Asia | 5 | 3% | 12 | 7% | 10 | 6% | 2 | 1% | 17 | 10% | 8 | 4% | 2 | 1% | 10 | 5% | 2.0 |
| Sub-Saharan Africa | 8 | 5% | 24 | 14% | 16 | 10% | 8 | 5% | 32 | 19% | 14 | 8% | 8 | 4% | 22 | 12% | 2.8 |
| **Total** | **56** | **33%** | **112** | **67%** | **80** | **48%** | **32** | **19%** | **168** | **100%** | **150** | **82%** | **33** | **18%** | **183** | **100%** | **3.3** |

## Network size

The 56 egos identified 112 unique alters (80 treatment centers and 32 'other' organizations) for a total network representing 168 nodes (Table 1). Fifty of the 80 (63%) unique treatment centers identified by egos were listed on www.1rbw.org at the time of the study. Thus 62% (106/170) of 1RBW was represented in this study (Table 1). The most represented geographical regions were North America (35/168, 21%), East Asia and Pacific (33/168, 20%), Europe and Central Asia (33/168, 20%) and Sub-Saharan Africa (32/168, 19%) (Table 1). Most institutions were in middle-income countries (MICs) (51%, 85/168) and high-income countries (HICs) (72/168, 43%) (Table 2).

## Network collaborations

The 56 egos collectively identified 183 collaborations (Table 1). Collaborations between retinoblastoma treatment centers accounted for 82% (150/183). Only 14 (8%) indicated there was a Memorandum of Understanding (MoU) established to govern the relationship; 1 indicated there was none (1%); and 168 (92%) declined to answer.

The mean number of alters identified per ego was 3.3 (Table 1). North American nodes identified the highest mean alters per ego at 6.8, while Middle Eastern and North African egos reported the least, at 1 (Table 1). Nodes in HICs had the highest mean alters per ego at 5.1, followed by 3.0 in low-income countries (LICs) and 2.3 in MICs (Table 2).

A network diagram was generated based on these findings, depicting 9 distinct groups: one large (>10 nodes) interconnected group, with representation from all geographic regions and income settings; five small (<10 nodes) groups, consisting of nodes/alters from 1–4 geographic

**Table 2. Nodes, alters and ties by World Bank Country income status.**

| World Bank Country Income Status | Nodes | | Alters | | | | | | All Network Members | | Ties (as identified by Nodes) | | | | | | | |
|---|---|---|---|---|---|---|---|---|---|---|---|---|---|---|---|---|---|---|
| | | | Unique Alters | | Unique 'Treatment Center' Alters | | Unique 'Other' Alters | | | | Ties with 'Treatment Center' Alters | | Ties with "Other" Alters | | Total Ties | | Mean # of ties per node |
| | n | % | n | % | n | % | n | % | n | % | n | % | n | % | n | % | |
| Low | 3 | 2% | 8 | 5% | 6 | 4% | 2 | 1% | 11 | 7% | 7 | 4% | 2 | 1% | 9 | 5% | 3.0 |
| Middle | 34 | 20% | 51 | 30% | 40 | 24% | 11 | 7% | 85 | 51% | 66 | 36% | 11 | 6% | 77 | 42% | 2.3 |
| High | 19 | 11% | 53 | 32% | 34 | 20% | 19 | 11% | 72 | 43% | 77 | 42% | 20 | 11% | 97 | 53% | 5.1 |
| **Total** | **56** | **33%** | **112** | **67%** | **80** | **48%** | **32** | **19%** | **168** | **100%** | **150** | **82%** | **33** | **18%** | **183** | **100%** | **3.3** |

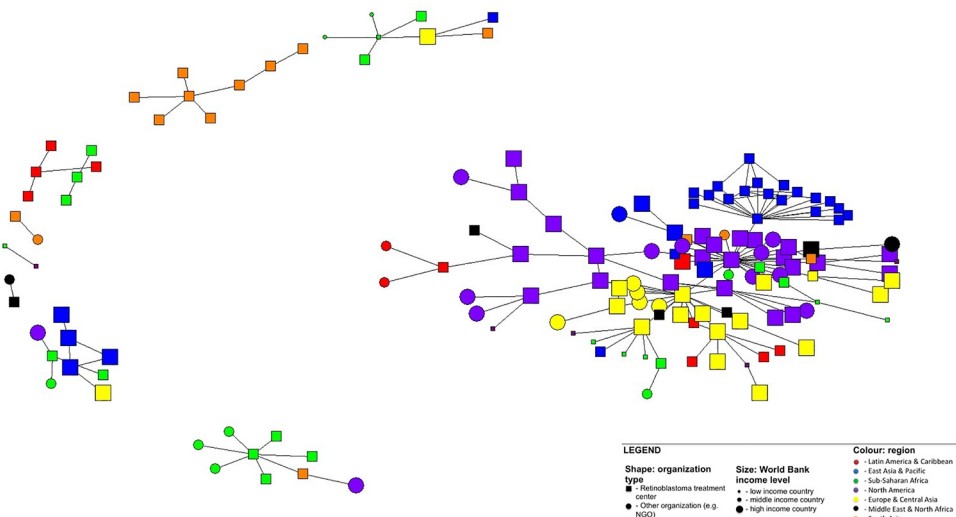

**Fig 1. Network diagram for all collaborative activities.** Network diagram depicting 56 egos and 112 unique alters connected by 183 collaborative activities (ties).

regions and >1 income setting; and three dyads (i.e. 2 nodes) each representing a single geographic region and a single income setting (Fig 1). Overall network density was 1.3% (S5 File).

Measures of centrality generally indicated that larger treatment centers in HICs were most central to networks of overall collaboration, sharing of information and tangible resources, joint planning, and patient referrals. Three centers, in particular, were most central to the network of overall partnership, one in North America (normalized degree centrality of 19.5%), one in Europe (8.4%), and one in East Asia (11.0%), each representing large tertiary medical centers with well-established expertise in retinoblastoma. Interestingly, the center in East Asia had nearly equal measure of in- and out-degree centrality (11.0% versus 10.4$), whereas the European and North American centers had much larger out-degree centrality relative to much smaller in-degree centrality (8.4% versus 3.2% and 19.5% versus 3.2%, respectively).

These three centers also had similar dominance in degree and betweenness centrality, relative to other centers within the network. This suggests that not only are these three centers highly connected to other centers, but that they serve as part of the most direct paths of connections amongst other centers. This implies a network structure where additional cross-linkages among centers is possible (and possibly desired), as these three centers serve in critical positions for multiple other centers in order to be connected with other centers in the network.

## Collaborative activities

Patient referrals were the most commonly reported collaboration, followed by information sharing, patient consultation, research, resource sharing, twinning/capacity building, joint planning and other. Network density statistics are reported in S5 File, and network diagrams for each activity are shown in Fig 2 and S6 File. The majority of patient referrals were between treatment centers, however there were rare instances of referrals being received or sent to other organizations (presumably to be further referred to a treatment facility). Information shared was typically in the form of treatment protocols, scientific publications, clinical care guidelines, conference announcements, funding opportunities and patient information for continuation of care. Patient consultation collaborations were established as early as 1985, and

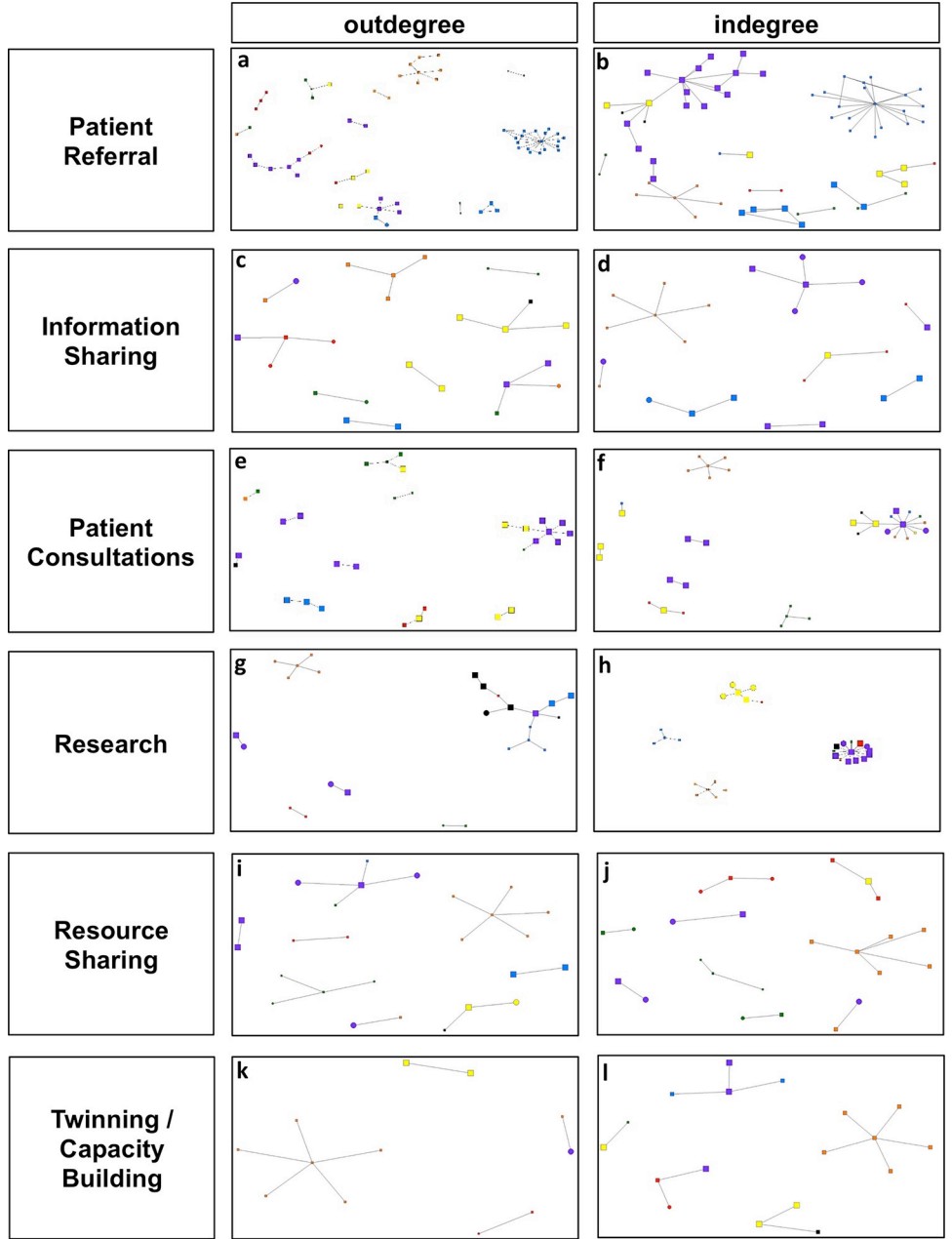

**Fig 2. Network diagram for outdegree and indegree collaborative activities.** Network diagrams depicting (a) outdegree patient referral, (b) indegree patient referral, (c) outdegree information sharing, (d) indegree information sharing, (e) outdegree information sharing, (f) indegree patient consultation, (g) outdegree patient consultation, (h) indegree research, (i) outdegree research, (j) indegree research sharing, (k) outdegree twining/capacity building and (l) indegree twining/capacity building.

as recently as 2014. Some research collaborations were established as early as 1996, but most reported a start date of 2005 or later. Resources shared included retinoblastoma cell lines, equipment, educational resources for patients and funding for participation in international meetings or development of guidelines. Twinning/capacity building initiatives were initiated in 2005 or later. Joint planning activities consisted of developing treatment protocols, conference organization, development of a virtual consultation system, design of capacity building

initiatives (including resource planning and training), developing a national strategy for retinoblastoma and planning awareness campaigns. Other activities included serving as adjunct faculty, developing clinical care innovations, fundraising for patient treatment, conducting a national tumor board, mentoring trainees, establishing links between partners and giving invited lectures.

### Thematic analysis

Four themes emerged from the semi-structured interviews aiming to elucidate the features of the partnerships and collaborations identified in the analysis (S7 File).

**1. Conceptualization of partnership.** No uniform description of 'partnership' emerged from discussions with key informants. Where some described informal connections between colleagues who called upon each other for various activities, others described formal partnerships guided by regular meetings, institutional support and contracts with clearly stated goals. Diverse thoughts on leadership also emerged; some key informants valued equality among partners, while others suggested that clear leadership by one partner, particularly those with resources and expertise, was required for success. Finally, there were differences related to the preferred 'closeness' of ties. Some key informants suggested that partnerships were best served by close personal connections and even friendships, others saw benefit even in weak ties, noting that these types of connections could still yield valuable opportunities for knowledge exchange and transfer.

**2. Primary motivation for collaboration.** The primary purpose of partnership was to improve patient outcomes. Improvements in capacity to treat, and improve retinoblastoma survival, were widely recognized as the most common benefits of partnership.

**3. Common challenges to collaboration.** Maintaining ties between partners in disparate settings and capacities was difficult, particularly when it came to the process of procuring financial resources to maintain joint projects, learning to work in a different health care context, and experiences with political instability that threatened to interfere with activities. Another common challenge identified was related to managing interpersonal relationships. However, most centers also emphasized that ensuring consistent and clear communication between partners was beneficial in identifying and mitigating communication challenges.

**4. Benefits of partnership.** Key informants from HICs often perceived the main benefits of partnership for their own centers as instrumental to improving research capacity, but commonly did not see this as a priority for LMICs. Instead they noted the benefits of using their resources and expertise to treat patients or build treatment capacity in the LMIC partner's setting. Similarly, key informants from LMICs saw the benefits of partnerships with well-resourced, expert HIC centers as a means to improve local capacity and physician knowledge. Unlike the HIC key informants however, the LMIC key informants often did cite the importance of research as a benefit of partnerships. They additionally recognized that partnerships with local institutions, or institutions working in similar contexts to their own, could offer more benefits to addressing the challenges associated with retinoblastoma. In fact, in one example, a key informant in Africa noted that their activities with an international NGO and treatment center were instrumental to their initial training and education, and served as an initial catalyst, empowering them to engage new local partners to manage retinoblastoma in their country.

## Discussion

Global cooperation, collaboration and partnership are endorsed as means to solve complex global health challenges, such as the achieving the Sustainable Development Goals [25]. Yet,

little is known about how collaborations are structured and work to meet their intended goals. We aimed to characterize the global collaborations working to improve survival for retinoblastoma. We focused our analysis on the 1RBW network, which is documented in a publically available database (www.1rbw.org). The 1RBW network is inclusive of the majority of global ophthalmic treatment centers, with countries lacking representation in the network expected to manage just 7% of the world's estimated retinoblastoma population (www.1rbw.org).

Our network analysis revealed that the global retinoblastoma community consists of a large group of actors located in all World Bank regions and income settings (Table 1). The identified alters were largely already members of the 1RBW network of treatment centers (Table 1). Alters also included other types of organizations, such as advocacy organizations and charities, which participated in all types of collaborations (Fig 2), underscoring their important role in the global retinoblastoma community.

Key informants reported that partnerships emerged out of a desire to improve the health and wellbeing of patients; in practice, this was most often translated into collaborative activities centered on delivery of healthcare, namely patient referrals, information sharing (i.e. treatment protocols, publications) and consultations (Fig 2). These types of activities were also the most longstanding collaborative activities reported, arguably consistent with the historic 'international health' approach focused on providing clinical care to vulnerable populations, compared to its successor, 'global health', which is focused on reducing health inequities and involves disciplines beyond the health sciences [1]. Consistent with the latter approach, the recent Lancet Commission on Global Eye Health posits that improving eye health is key to achieving the Sustainable Development Goals, framing it as a key global health issue [26]. The Commission emphasizes the importance of improving not just capacity, accessibility and quality of eye health services, but also the vital need for reliable data and innovation to drive progress [26]. Twinning/capacity building and joint planning activities reported in this study are most aligned with this model, as well as calls to improve human resources for eye health in LMICs [27], but represented the least common activities in 1RBW. Arguably, the majority of collaborations reported in this study could benefit from aligning their activities with a global health approach to more sustainably achieve their goals.

The 183 collaborations represented a network density of 1.3%, suggesting that much more inter-collaboration could be taking place within the network. The network diagram depicts this low network density visually, revealing very few well-connected actors at the center of collaborative activities, with most nodes situated in the outer peripheries or completely detached from the main network (Fig 1). Connectivity of nodes is an approximate measure of their power and strength within networks; for 1RBW, the most well-connected nodes were located in North America and Europe (Table 1) or HICs (Table 2). Further, centrality measures from the most well-connected centers showing greater unidirectional out-degree interaction with much less in-degree interaction was suggestive of a power imbalance; this was not observed with a well-connected center in East Asia, which demonstrated bidirectional interactions of equivalent centrality. Coupled with this relatively large degree centrality belonging to just a few centers, was a parallel concentration of betweenness centrality, indicative of the position these centers have in line with connections between other, less connected centers. Thus, centers may be relying on one of these three large and connected centers to connect with other, less connected centers within their region. This exposes a striking irony: while the primary motivation behind collaboration was to promote patient survival, the apparent dominance of actors in HICs might inadvertently promote the very structural inequities that contribute to the poor outcomes observed for retinoblastoma in LMICs. Poorly formed global health partnerships can have serious consequences, such as propagating paternalistic and neocolonial attitudes and practices, to destabilizing national health care systems [28–30]. In light of the

recent movement to decolonize global health [31], it is an opportune time for the global retinoblastoma community to acknowledge this power imbalance, and strive towards shifting the paradigm on which collaborations are currently modeled.

The interviews spoke to the challenges in forming partnerships among actors in disparate income settings, and suggested that frequent and open communication was key to mitigating these challenges. However, key informants did not comment on how equality or mutual communication could be established or determined. Furthermore, collaborative activities were not commonly governed by MoUs. Formal contracts such as MoUs can promote transparency and accountability in the establishment of roles, responsibilities, goals and sharing of benefits, and are recommended for partnerships between privileged and non-privileged actors [32]. Routine evaluation of partnerships with the use of formal assessment tools could promote communication, serve as indicators of partnership health, and identify approaches to promote equitable relationships [33].

Our study uncovered contrasting views on the placement of collaborative research activities in LMICs. There was a prevailing perception among key informants in HICs that research was not a priority for LMICs. However, key informants from LMICs pointed to a key role for international partnerships in improving research capacity. The views of HIC key informants may reflect paternalistic attitudes prevalent in international research partnerships [30], and should be challenged. Research has an important role in reducing global health inequities [1]; for retinoblastoma, scientific study is precisely what has contributed to the evidence-based guidelines [34] that result in good survival in HICs, and evaluation of capacity building initiatives that promote good outcomes in LMICs [12, 14, 35–37]. There is an impetus to conduct more retinoblastoma research in LMICs, as that is where the majority of children with retinoblastoma live; knowledge produced elsewhere may not necessarily be transferrable to their context. Clinical retinoblastoma research partnerships between HICs and LMICs tend to involve well-equipped academic centers in emerging economies (for example in India [38]). While such partnerships offer strategic benefits, they may exclude important patient populations, study contexts, and clinician perspectives that are held in less-resourced or less-connected centers. Further efforts to include treatment centers at the peripheries of the 1RBW network, strengthen research capacity, and generate research beyond the clinical (e.g. basic science, implementation research) could result in new knowledge that leads to more widespread patient benefit. In the time since our network analysis was completed, the 1RBW network map served to reveal potential investigators who were subsequently invited to join a collaborative study of global retinoblastoma presentation [39]. This example of global coordination and cooperation to complete a retrospective study was the largest effort of its kind for retinoblastoma, and holds promise for further collaboration to conduct additional novel and robust studies in future. However, an examination of the investigators who declined to collaborate in this study, and their reasons why, could reveal important structural inequities that still need to be addressed in order to facilitate an inclusive environment for research participation and leadership.

Furthermore, though it should go without saying, retinoblastoma research partnerships do not necessarily have to involve partners from HICs. Interestingly, while nodes in Latin America and the Caribbean were relatively underrepresented in our study, the literature demonstrates successful clinical research collaborations for retinoblastoma resulting from the Central American Association of Pediatric Hematology Oncology (AHOPCA) [7, 8] and the Latin American Pediatric Oncology Group (GALOP) [9]. Such 'South-South' collaborations may be more successful due to the fact that partners experience similar challenges, contexts and motivation to find a common solution [40], a view revealed in our interviews. Indeed, the highly dense network we identified in East Asia, formed mainly of retinoblastoma treatment centers in China, may be reflective of a such a challenge addressed by a homegrown solution; specifically, a model of shared care was implemented to eliminate the financial and psychosocial

burden on patients requiring referral, by instead requiring the medical staff to move between treatment centers [41].

A limitation of this study was that just 56 of the 170 centers (33%) listed on *1RBW* participated in the study, and we did not perform snowball sampling of identified alters outside of *1RBW*, meaning that our analysis may have missed relevant collaborative activities. Given that the study was available only in the English language, it is possible that some 1RBW members may have chosen not to participate due to language barriers. However, our data still implicated 68% of network members (Table 1), a significant proportion of the overall network. In addition, since participants were asked to report on activities in the last 12 months but were not asked to provide an end date (if relevant), it is possible that some activities represented one-time activities rather than ongoing collaborations. The rather uniform positionality of the authors is also a limitation of this study; inclusion of more diverse study team members (e.g. representatives of LMICs or secondary retinoblastoma centers) or conducting the study in additional languages may have yielded richer insights.

In summary, the findings suggest that there exists a large group of actors working to reduce retinoblastoma mortality, but there is room to expand connectivity and interaction through active efforts to include actors located at network peripheries. This can be achieved by dissemination of this and other similar work focused on achieving representation and equity in international partnerships [42], which could stimulate important discussion, knowledge exchange and action in global retinoblastoma activities. Professional bodies in pediatric and ocular oncology could play a role in such efforts, for example by inviting and supporting underrepresented actors to attend international conferences to increase their visibility and share their work. It has become increasingly common for retinoblastoma conference sessions to highlight global retinoblastoma partnerships, however, they still remain largely led by HIC speakers focused on clinical impact of activities, with little attention paid to how partnerships are developed, structured and maintained. Professional bodies can support the latter by making it a requirement to include a statement of reflexivity with respect to international partnerships, as has been suggested for research publications involving the same [43]. Furthermore, additional research, such as an examination of how participation in certain categories of collaboration within 1RBW are related to participation (or lack thereof) in other categories of collaboration, may help better elucidate the overall dynamics of this large collaborative. Going forward, we recommend that members of the global retinoblastoma community are made aware of power imbalances inherent within the existing network, and make concerted effort to research and re-design partnership activities to avoid perpetuating structural and social inequities.

## Supporting information

**S1 File. STROBE checklist for cross-sectional studies.** The "STrengthening of the Reporting of Observational studies in Epidemiology" checklist was applied to the reporting of this study. (DOCX)

**S2 File. Network analysis survey.** The questionnaire used in this study. (DOCX)

**S3 File. Semi-structured interview guide.** The semi-structured interview guide used in this study. (DOCX)

**S4 File. Egos by region and country.** Summary data of egos represented in this study, presented by geographical region and country. (DOCX)

**S5 File. Network density.** Summary network density statistics by overall and individual (in-degree and out-degree, where relevant) interactions.
(DOCX)

**S6 File. Network diagrams for joint planning and other activities.** Network diagrams depicting joint planning and other activities.
(DOCX)

**S7 File. Themes and sample quotes.** Resulting themes and supporting participant quotes from the qualitative analysis.
(DOCX)

**S8 File. Dataset.** The dataset used in this analysis.
(XLSX)

## Acknowledgments

The authors wish to thank members of *One Retinoblastoma World* who took the time to participate in the study.

## Author Contributions

**Conceptualization:** Kaitlyn Flegg, John Prochaska, Helen Dimaras.

**Data curation:** Hannah Girdler, Helen Dimaras.

**Formal analysis:** Hannah Girdler, Kaitlyn Flegg, John Prochaska, Helen Dimaras.

**Funding acquisition:** Helen Dimaras.

**Investigation:** Hannah Girdler, Kaitlyn Flegg, John Prochaska, Helen Dimaras.

**Methodology:** Hannah Girdler, Kaitlyn Flegg, John Prochaska, Helen Dimaras.

**Project administration:** Kaitlyn Flegg, Helen Dimaras.

**Resources:** Helen Dimaras.

**Software:** John Prochaska, Helen Dimaras.

**Supervision:** John Prochaska, Helen Dimaras.

**Validation:** Helen Dimaras.

**Visualization:** Hannah Girdler, John Prochaska, Helen Dimaras.

**Writing – original draft:** Hannah Girdler.

**Writing – review & editing:** Hannah Girdler, Kaitlyn Flegg, John Prochaska, Helen Dimaras.

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
