## [Decision Letter · Decision Letter 0]

1 Aug 2021

 PGPH-D-21-00219 Characterization of international partnerships in global retinoblastoma care and research: A network analysis. PLOS Global Public Health

Dear Dr. Dimaras,

Thank you for submitting your manuscript to PLOS Global Public Health. After careful consideration, we feel that it has merit but does not fully meet PLOS Global Public Health’s publication criteria as it currently stands. Therefore, we invite you to submit a revised version of the manuscript that addresses the points raised during the review process. 

We look forward to receiving your revised manuscript.

Kind regards,

Academic Editor

Journal Requirements:

Additional Editor Comments (if provided):

Please improve your paper by addressing the issues raised by the reviewers.

Ensure that you respond to all points and resubmit your paper for further consideration.

Please make sure to comply strictly with the Plos Global Public Health submission guidelines including the style and format.

Ensure that the paper references are up to date and the links are still valid.

Reviewer 1:

Thanks for the opportunity to review such an interesting piece of work. The authors did an excellent job in collecting valuable data on a relevant global health topic, which is eye care (or more specifically retinoblastoma care). The introduction, the qualitative analysis and the discussion are very stimulating.

The network analysis is also potentially very rich and interesting. However, I believe the authors fail to make a sufficiently good use of the data available in terms of network statistics. Additionally, the integration between the qualitative and the quantitative components of the study could be made more explicit in the discussion. I believe that conducting and reporting a few additional standard network analysis procedure would offer additional insights that can then feed into the discussion, adding interesting content and linking nicely the qual and quant components of this mixed methods paper.

Below a few specific comments on different paper sections:

Methodology

- Add network questionnaire in Appendix

- For the network analysis methods, an accessible “guide” is provided in Blanchet & James (2012), https://doi.org/10.1093/heapol/czr055. It is worth having a look (and potentially referencing) if you didn’t see it earlier.

- How did you choose which organisations (egos) to invite for the interview, of those who replied to the quant survey?

Results

- The results offer a valid description of the data. However, as it is, the paper fails to exploit the rich and interesting data that and most of all fails to integrate properly the “network analysis” with the qualitative analysis (see reference above, Blanchet & James 2012). Two low hanging fruits directly from the network analysis toolbox that could provide interesting insights for the discussion are:

1. Explore a few centrality measures, discussing the characteristic of the most central organizations in the networks and sub-networks identified by the study: who are these powerful organisations? What is their role? How does this affect the overall functioning of the network?

2. Potentially, it could be interesting explore the relationship between separate collaborative activities (which I assume are asked non-exclusively in the questionnaire). This could be explored graphically or – in a more straightforward way - more analytically, in terms of association between different networks of collaborative activities. This latter point could be addressed with simple adjacency matrix correlations, or using a more involved inferential approach (e.g. Exponential Random Graph Models). Questions that could be addressed with such tools are, for example: “How well does having information sharing predict having a patient referral network?”, or “Does sharing resources predict having patient shared consultation?”

- If the authors decide to add some of these points, the Methods section would need updating, accordingly

- Tables 1 and 3 could be united in a single Table, leaving room for some of the analyses above. The same holds true for Tables 2 and 4.

Discussion

- The discussion seems comprehensive and covers a lot of topics – It definitely shows that the authors are well aware of the challenges in this field

- What’s missing (in the discussion and also potentially in the Introduction) is a clear statement on how representative the 1RBW network is of the global eye health landscape, which would tell the reader how relevant these findings are for global health policies

- A mention to (and a contrast with) the goals and findings of initiatives such as the Lancet Global Health Commission on Global Eye Health seems necessary (e.g. https://doi.org/10.1016/S2214-109X(20)30488-5), yet missing

- Perhaps worth expanding/aligning to new potential results, if some of the additional points are addressed based on the comments above

- I found that two references (among others which the authors may know about) are missing and could add value to the paper if discussed in relation to the study results

1. https://doi.org/10.1186/1478-4491-12-44

2. https://doi.org/10.1093/heapol/czs031

Reviewer 2

Global burden of retinoblastoma is in LMICs, where there is also a high mortality rate despite it being a curable condition. International collaborations through 1RBW aim to support optimal care and improve diagnosis and outcomes for retinoblastoma. This paper aims to explore the structure of global collaborations for retinoblastoma and how this can contribute to improved outcomes in order to identify where collaboration can be strengthened and replicated. This is a novel approach to exploring how collaboration can be strengthened for health outcomes, and could be applied to networks beyond retinoblastoma.

Introduction

Reference 4 weblink no longer correct

It would be helpful if lines 90-91 could be elaborated slightly to provide some examples of how global collaboration can improve retinoblastoma outcomes (or of how such collaboration has improved outcomes in other conditions). Clearly this is one of the aims of the paper, but ten references have been given for collaborations that have shown progress towards improving outcomes – a brief summary here would be useful.

Methods

It would be useful to have some further details on the sample. E.g., are the 1RBW centres all of similar size, are they urban/rural? And for the results, how do those who responded to the survey compare with those who did not? As it’s quite a small sample size this information would be helpful in thinking about whether the results are likely to be similar for other centres. There are a lot more respondents in China than elsewhere; is this related to the number of centres there or was there a particular effort to recruit there? Some treatment centres are not in 1RBW; how do centres get involved in that, does 1RBW represent the main centres in each country?

Were respondents given a time frame for any collaborating activity (or just asked about start dates on ongoing activities)? Did you have any information on the person completing the survey (and did this make a difference to responses)?

Semi-structured interview sample – does ‘extent of connectivity’ mean that participants were recruited based on having few to many connections, or did this take into account their organisations’ centrality? A bit more detail on sampling would be helpful here as well – how were the 23 chosen?

The topic guide focuses on asking about multiple partnerships (strongest partners/weakest partnerships), but little on the barriers to achieving partnership in the first place. Many of the egos only have one alter, so some of these questions seem like they’d be difficult to answer. Were participants only recruited if they had multiple partnerships?

It should be included in the methods that all interviews conducted in English. Perhaps more should be said about this in the limitations, particularly given the message of the paper being about the need for ‘active efforts to include actors located at network peripheries’ and that ‘members of the retinoblastoma community remain aware of power imbalances’ – is there some irony in this message given that those at the periphery may have been less able to be involved in the study? Do researchers see themselves as part of this network, and if so by only conducting the research in English does this go against the very messages coming out of the study?

Reciprocity of relationships was assumed, but surely this could have been checked since many of the reported partnerships were others in the sample?

Results

Although 62% of 1RBW are represented in the study, some of these did not actually provide information on their collaborations (& reciprocity was assumed). Though I note this is mentioned in the limitations section.

Figure 1 – it’s difficult to differentiate between node size; could you make the bigger slightly bigger?

It’s not clear from Figure 1 which egos had responded to the survey and which were only identified by others. That there was no snowball sampling from responses should be highlighted as a limitation in the paper (it does not attempt to reflect a whole network), and a note should be added to Figure 1 to highlight that some centres/organisations have fewer collaborations than likely in reality because they did not provide information about their collaborations.

Thematic analysis provides a fairly descriptive overview of the themes rather than really going into any depth about participants’ views on the issues that are raised in the discussion. Given that the discussion goes into some detail on issues of power in the network, it would have been interesting to have presented some more in-depth findings related to this (from different veiwpoints) in the results. As it is, the results section provides a summary of interviews rather than any analysis backed up with evidence (e.g., quotes, though I note these are in the supplementary file, if word count allowed it would have been good to see some of these making up the main points in the findings section).

Discussion

The authors state that connectivity of nodes is an approximate measure of their power and strength within networks, yet this does not appear to be something that was asked about in the interviews (or at least not reported in much detail). This is unfortunate as the interviews were conducted after the survey and would have been expected to probe into the relationships uncovered in the network analysis.

This is an interesting discussion around the issues touched upon in the findings, but could be expanded to provide more detail on how this research will be taken forward; how will the research contribute to “strive towards shifting the paradigm on which collaborations are currently modelled”. The dissemination of the work could support the recommendations (for organisations to be aware of power imbalances within the existing network, the need for effort to re-design partnership activities to avoid perpetuating structural and social inequities). E.g., What could you do with the network analysis? Are there any network strengthening interventions, or dissemination activities planned with 1RBW to determine how to use these findings to support collaboration? It’s not clear if the collaborative study that has started since this network analysis drew upon these findings at all (line 308). From the survey it is clear that there are organisations not in 1RBW but are nonetheless important members of the community. Does the network need to be expanded in some way?

Reviewers' comments:

Reviewer's Responses to Questions

**Comments to the Author**

1. Does this manuscript meet PLOS Global Public Health’s publication criteria? Is the manuscript technically sound, and do the data support the conclusions? The manuscript must describe methodologically and ethically rigorous research with conclusions that are appropriately drawn based on the data presented.

Reviewer #1: Yes

Reviewer #2: Yes

2. Has the statistical analysis been performed appropriately and rigorously?

Reviewer #1: Yes

Reviewer #2: I don't know

3. Have the authors made all data underlying the findings in their manuscript fully available (please refer to the Data Availability Statement at the start of the manuscript PDF file)?

Reviewer #1: No

Reviewer #2: No

4. Is the manuscript presented in an intelligible fashion and written in standard English?

Reviewer #1: Yes

Reviewer #2: Yes

5. Review Comments to the Author

Reviewer #1: Thanks for the opportunity to review such an interesting piece of work. The authors did an excellent job in collecting valuable data on a relevant global health topic, which is eye care (or more specifically retinoblastoma care). The introduction, the qualitative analysis and the discussion are very stimulating.

The network analysis is also potentially very rich and interesting. However, I believe the authors fail to make a sufficiently good use of the data available in terms of network statistics. Additionally, the integration between the qualitative and the quantitative components of the study could be made more explicit in the discussion. I believe that conducting and reporting a few additional standard network analysis procedure would offer additional insights that can then feed into the discussion, adding interesting content and linking nicely the qual and quant components of this mixed methods paper.

Below a few specific comments on different paper sections:

Methodology

- Add network questionnaire in Appendix

- For the network analysis methods, an accessible “guide” is provided in Blanchet & James (2012), https://doi.org/10.1093/heapol/czr055. It is worth having a look (and potentially referencing) if you didn’t see it earlier.

- How did you choose which organisations (egos) to invite for the interview, of those who replied to the quant survey?

Results

- The results offer a valid description of the data. However, as it is, the paper fails to exploit the rich and interesting data that and most of all fails to integrate properly the “network analysis” with the qualitative analysis (see reference above, Blanchet & James 2012). Two low hanging fruits directly from the network analysis toolbox that could provide interesting insights for the discussion are:

1. Explore a few centrality measures, discussing the characteristic of the most central organizations in the networks and sub-networks identified by the study: who are these powerful organisations? What is their role? How does this affect the overall functioning of the network?

2. Potentially, it could be interesting explore the relationship between separate collaborative activities (which I assume are asked non-exclusively in the questionnaire). This could be explored graphically or – in a more straightforward way - more analytically, in terms of association between different networks of collaborative activities. This latter point could be addressed with simple adjacency matrix correlations, or using a more involved inferential approach (e.g. Exponential Random Graph Models). Questions that could be addressed with such tools are, for example: “How well does having information sharing predict having a patient referral network?”, or “Does sharing resources predict having patient shared consultation?”

- If the authors decide to add some of these points, the Methods section would need updating, accordingly

- Tables 1 and 3 could be united in a single Table, leaving room for some of the analyses above. The same holds true for Tables 2 and 4.

Discussion

- The discussion seems comprehensive and covers a lot of topics – It definitely shows that the authors are well aware of the challenges in this field

- What’s missing (in the discussion and also potentially in the Introduction) is a clear statement on how representative the 1RBW network is of the global eye health landscape, which would tell the reader how relevant these findings are for global health policies

- A mention to (and a contrast with) the goals and findings of initiatives such as the Lancet Global Health Commission on Global Eye Health seems necessary (e.g. https://doi.org/10.1016/S2214-109X(20)30488-5), yet missing

- Perhaps worth expanding/aligning to new potential results, if some of the additional points are addressed based on the comments above

- I found that two references (among others which the authors may know about) are missing and could add value to the paper if discussed in relation to the study results

1. https://doi.org/10.1186/1478-4491-12-44

2. https://doi.org/10.1093/heapol/czs031

Reviewer #2: Global burden of retinoblastoma is in LMICs, where there is also a high mortality rate despite it being a curable condition. International collaborations through 1RBW aim to support optimal care and improve diagnosis and outcomes for retinoblastoma. This paper aims to explore the structure of global collaborations for retinoblastoma and how this can contribute to improved outcomes in order to identify where collaboration can be strengthened and replicated. This is a novel approach to exploring how collaboration can be strengthened for health outcomes, and could be applied to networks beyond retinoblastoma.

Introduction

Reference 4 weblink no longer correct

It would be helpful if lines 90-91 could be elaborated slightly to provide some examples of how global collaboration can improve retinoblastoma outcomes (or of how such collaboration has improved outcomes in other conditions). Clearly this is one of the aims of the paper, but ten references have been given for collaborations that have shown progress towards improving outcomes – a brief summary here would be useful.

Methods

It would be useful to have some further details on the sample. E.g., are the 1RBW centres all of similar size, are they urban/rural? And for the results, how do those who responded to the survey compare with those who did not? As it’s quite a small sample size this information would be helpful in thinking about whether the results are likely to be similar for other centres. There are a lot more respondents in China than elsewhere; is this related to the number of centres there or was there a particular effort to recruit there? Some treatment centres are not in 1RBW; how do centres get involved in that, does 1RBW represent the main centres in each country?

Were respondents given a time frame for any collaborating activity (or just asked about start dates on ongoing activities)? Did you have any information on the person completing the survey (and did this make a difference to responses)?

Semi-structured interview sample – does ‘extent of connectivity’ mean that participants were recruited based on having few to many connections, or did this take into account their organisations’ centrality? A bit more detail on sampling would be helpful here as well – how were the 23 chosen?

The topic guide focuses on asking about multiple partnerships (strongest partners/weakest partnerships), but little on the barriers to achieving partnership in the first place. Many of the egos only have one alter, so some of these questions seem like they’d be difficult to answer. Were participants only recruited if they had multiple partnerships?

It should be included in the methods that all interviews conducted in English. Perhaps more should be said about this in the limitations, particularly given the message of the paper being about the need for ‘active efforts to include actors located at network peripheries’ and that ‘members of the retinoblastoma community remain aware of power imbalances’ – is there some irony in this message given that those at the periphery may have been less able to be involved in the study? Do researchers see themselves as part of this network, and if so by only conducting the research in English does this go against the very messages coming out of the study?

Reciprocity of relationships was assumed, but surely this could have been checked since many of the reported partnerships were others in the sample?

Results

Although 62% of 1RBW are represented in the study, some of these did not actually provide information on their collaborations (& reciprocity was assumed). Though I note this is mentioned in the limitations section.

Figure 1 – it’s difficult to differentiate between node size; could you make the bigger slightly bigger?

It’s not clear from Figure 1 which egos had responded to the survey and which were only identified by others. That there was no snowball sampling from responses should be highlighted as a limitation in the paper (it does not attempt to reflect a whole network), and a note should be added to Figure 1 to highlight that some centres/organisations have fewer collaborations than likely in reality because they did not provide information about their collaborations.

Thematic analysis provides a fairly descriptive overview of the themes rather than really going into any depth about participants’ views on the issues that are raised in the discussion. Given that the discussion goes into some detail on issues of power in the network, it would have been interesting to have presented some more in-depth findings related to this (from different veiwpoints) in the results. As it is, the results section provides a summary of interviews rather than any analysis backed up with evidence (e.g., quotes, though I note these are in the supplementary file, if word count allowed it would have been good to see some of these making up the main points in the findings section).

Discussion

The authors state that connectivity of nodes is an approximate measure of their power and strength within networks, yet this does not appear to be something that was asked about in the interviews (or at least not reported in much detail). This is unfortunate as the interviews were conducted after the survey and would have been expected to probe into the relationships uncovered in the network analysis.

This is an interesting discussion around the issues touched upon in the findings, but could be expanded to provide more detail on how this research will be taken forward; how will the research contribute to “strive towards shifting the paradigm on which collaborations are currently modelled”. The dissemination of the work could support the recommendations (for organisations to be aware of power imbalances within the existing network, the need for effort to re-design partnership activities to avoid perpetuating structural and social inequities). E.g., What could you do with the network analysis? Are there any network strengthening interventions, or dissemination activities planned with 1RBW to determine how to use these findings to support collaboration? It’s not clear if the collaborative study that has started since this network analysis drew upon these findings at all (line 308). From the survey it is clear that there are organisations not in 1RBW but are nonetheless important members of the community. Does the network need to be expanded in some way?

6. PLOS authors have the option to publish the peer review history of their article (what does this mean?). If published, this will include your full peer review and any attached files.

**Do you want your identity to be public for this peer review?** For information about this choice, including consent withdrawal, please see our Privacy Policy.

Reviewer #1: No

Reviewer #2: No

---

## [Decision Letter · Decision Letter 1]

2 Nov 2021

PGPH-D-21-00219R1

Characterization of international partnerships in global retinoblastoma care and research: A network analysis.

Dear Dr. Dimaras,

Thank you for submitting your manuscript to PLOS Global Public Health. After careful consideration, we feel that it has merit but does not fully meet PLOS Global Public Health’s publication criteria as it currently stands. Therefore, we invite you to submit a revised version of the manuscript that addresses the points raised during the review process.

We look forward to receiving your revised manuscript.

Kind regards,

The Academic Editor

Journal Requirements:

Additional Editor Comments (if provided):

Reviewers' comments:

Reviewer's Responses to Questions

**Comments to the Author**

1. If the authors have adequately addressed your comments raised in a previous round of review and you feel that this manuscript is now acceptable for publication, you may indicate that here to bypass the “Comments to the Author” section, enter your conflict of interest statement in the “Confidential to Editor” section, and submit your "Accept" recommendation.

Reviewer #1: (No Response)

Reviewer #2: (No Response)

2. Does this manuscript meet PLOS Global Public Health’s publication criteria? Is the manuscript technically sound, and do the data support the conclusions? The manuscript must describe methodologically and ethically rigorous research with conclusions that are appropriately drawn based on the data presented.

Reviewer #1: Yes

Reviewer #2: Yes

3. Has the statistical analysis been performed appropriately and rigorously?

Reviewer #1: Yes

Reviewer #2: I don't know

4. Have the authors made all data underlying the findings in their manuscript fully available (please refer to the Data Availability Statement at the start of the manuscript PDF file)?

Reviewer #1: Yes

Reviewer #2: Yes

5. Is the manuscript presented in an intelligible fashion and written in standard English?

Reviewer #1: Yes

Reviewer #2: Yes

6. Review Comments to the Author

Reviewer #1: The authors made an excellent work in addressing comments to the previous version of the manuscript. The paper is vastly improved and will make a great contribution to the literature. I only have two last minor comments, related to the centrality measures introduced. Firstly, can the authors report the characteristics of the most central nodes (centrality measure value, geographical region, type of organisation, etc) in terms of degree and betweenness centrality? Secondly, do centrality rankings differ across centrality measures? What does that imply? Since you computed it, why don't you explicitly discuss betweenness centrality? I believe addressing these points would allow an extra step up in terms of interpreting the eyecare partnershipsrepresented, and at the same time make an even better use of the rich network data collected.

Reviewer #2: Thank you for responding to comments and making edits to the paper, which have been very helpful to my understanding of the work. I think most have been addressed but there are a few outstanding points/questions:

1. Regarding the response to the question that I previously asked about the number of respondents in China (from supplementary file S3), and that participants were eligible if they were a healthcare professional (e.g. oncologist, ophthalmologist) leading a retinoblastoma team at a treatment centre, does this mean that one ophthalmologist responded to the survey 18 times? If so, I think this is important to note, given the implications (particularly that there is a link between each of these centres as there is one ophthalmologist travelling between them).

2. Regarding the point about the time frame for collaborating activity – the authors’ response indicates that the activity could be historical and perhaps not reflect an ongoing collaboration between centres. This may be considered a limitation of the survey, and should be noted in the paper.

3. The comments about the results section providing a descriptive overview of themes, and about the lack of information from the interviews about power issues (which was identified and highlighted in the discussion) remain. The additional information on centrality helps, but it is a limitation that the qualitative interviews and analysis did not delve into this issue, particularly given the focus given to it in the discussion.

7. PLOS authors have the option to publish the peer review history of their article (what does this mean?). If published, this will include your full peer review and any attached files.

**Do you want your identity to be public for this peer review?** For information about this choice, including consent withdrawal, please see our Privacy Policy.

Reviewer #1: No

Reviewer #2: No

---

## [Editor Report · Decision Letter 2]

24 Nov 2021

Characterization of international partnerships in global retinoblastoma care and research: A network analysis.

PGPH-D-21-00219R2

Dear Dr. Dimaras,

We're pleased to inform you that your manuscript has been judged scientifically suitable for publication and will be formally accepted for publication once it meets all outstanding technical requirements.

Within one week, you'll receive an e-mail detailing the required amendments. When these have been addressed, you'll receive a formal acceptance letter and your manuscript will be scheduled for publication.

An invoice for payment will follow shortly after the formal acceptance. To ensure an efficient process, please log into Editorial Manager at https://www.editorialmanager.com/pgph/ click the 'Update My Information' link at the top of the page, and double check that your user information is up-to-date. If you have any billing related questions, please contact our Author Billing department directly at authorbilling@plos.org.

Kind regards,

The Academic Editor